# Could Iron-Nitrogen Doping Modulate the Cytotoxicity of TiO_2_ Nanoparticles?

**DOI:** 10.3390/nano12050770

**Published:** 2022-02-25

**Authors:** Ionela Cristina Nica, Bogdan Andrei Miu, Miruna S. Stan, Lucian Diamandescu, Anca Dinischiotu

**Affiliations:** 1Department of Biochemistry and Molecular Biology, Faculty of Biology, University of Bucharest, 91-95 Splaiul Independentei, 050095 Bucharest, Romania; cristina.nica@drd.unibuc.ro (I.C.N.); miu.bogdan-andrei@s.bio.unibuc.ro (B.A.M.); anca.dinischiotu@bio.unibuc.ro (A.D.); 2Research Institute of the University of Bucharest—ICUB, University of Bucharest, 050657 Bucharest, Romania; 3National Institute of Materials Physics (NIMP), Atomistilor 405A, Magurele, 077125 Bucharest, Romania; diamand@infim.ro

**Keywords:** titanium dioxide, P25 Degussa nanoparticles, iron doping, photocatalysts, human cytotoxicity

## Abstract

Titanium dioxide nanoparticles (TiO_2_ NPs) are found in several products on the market that include paints, smart textiles, cosmetics and food products. Besides these, TiO_2_ NPs are intensively researched for their use in biomedicine, agriculture or installations to produce energy. Taking into account that several risks have been associated with the use of TiO_2_ NPs, our aim was to provide TiO_2_ NPs with improved qualities and lower toxicity to humans and the environment. Pure TiO_2_ P25 NPs and the same NPs co-doped with iron (1%) and nitrogen atoms (P25-Fe(1%)-N NPs) by hydrothermal treatment to increase the photocatalytic activity in the visible light spectrum were in vitro evaluated in the presence of human lung cells. After 24 and 72 h of incubation, the oxidative stress was initiated in a time- and dose-dependent manner with major differences between pure P25 and P25-Fe(1%)-N NPs as revealed by malondialdehyde and reactive oxygen species levels. Additionally, a lower dynamic of autophagic vacuoles formation was observed in cells exposed to Fe-N-doped P25 NPs compared to the pure ones. Therefore, our results suggest that Fe-N doping of TiO_2_ NPs can represent a valuable alternative to the conventional P25 Degussa particles in industrial and medical applications.

## 1. Introduction

Metal oxide nanoparticles (NPs) display a high ability for various applications including microelectronics, energy storage, environmental decontamination, gas sensing, ceramic fabrication, biomedicine (imaging, drug delivery, therapy and theranostics) [1,2,3]. Titanium dioxide (TiO_2_) NPs are one of the most widely used nanoscale materials in several market products that contain TiO_2_. These include paints [4], smart textiles [5], cosmetics, skin care [6] or food products [7]. Recently, the entire field of nanotechnology has made considerable progress, and TiO_2_ NPs were introduced in biomedicine [8], agriculture [9] and electricity [10] and hydrogen [11] production.

Being so widespread in consumer products, it is necessary to assess the risks that TiO_2_ presents to human health. The accumulation over time of several scientific data supporting the carcinogenic effect of TiO_2_ in nanometric form has led the International Agency for Research on Cancer (IARC) to classify this material in group 2B carcinogens in 2010, this characteristic being attributed to substances that possibly trigger the development of malignant tumors in the human body. However, the scientific evidence was not convincing or sufficient since this agency’s report largely refers to TiO_2_ NPs that come into contact with humans through occupational inhalation [12]. Moreover, since 2009, at the level of the European Union, the concentration of TiO_2_ used in cosmetics is restricted to a percentage of 25% of product [13]. Ten years later, a new condition of use was introduced, prohibiting certain cosmetic preparations that can lead to the inhalation of TiO_2_ NPs [13]. The most recent regulation since 2020 is that the European Commission officially classified TiO_2_ as a category 2 suspected carcinogen by inhalation [14].

Despite all these facts, the academic community has not yet reached a consensus on the negative effects of NPs on health, with some authors suggesting that more scientific data are still needed [15,16,17]. However, it is currently accepted that nanomaterials can generate oxidative stress that is involved in triggering proinflammatory or apoptotic signaling pathways at a cellular level [18]. Numerous studies reported that TiO_2_ NPs induced a decrease in cell viability, while other in vitro studies showed that exposure to TiO_2_ NPs can also trigger apoptotic processes in BEAS-2B lung epithelial cells [19], WI38 lung fibroblasts, MCF10A non-malignant breast epithelial cells [20] and 16HBE14o bronchial epithelial cells [21]. In addition, it was reported that TiO_2_ NPs can induce cell death in a p53- [22] and Bax/Bcl-2-independent manner [23].

Nanotechnology can further advance only after all risks associated with NP use are clearly identified. Therefore, our aim was to prepare TiO_2_ NPs with improved qualities and lower toxicity to humans and the environment. Iron-nitrogen (Fe-N) doping of TiO_2_ NPs can represent a valuable alternative to the conventional P25 Degussa particles in industrial and medical applications.

## 2. Materials and Methods

### 2.1. Synthesis and Physicochemical Characterization of TiO_2_ NPs

TiO_2_ Degussa P25 powder (Sigma-Aldrich, St. Louis, MO, USA) was impregnated with 1% Fe and N atoms as follows: adequate amounts of TiO_2_ (Aeroxide^®^ P25), FeCl_3_·6H_2_O and urea were dispersed under mechanical and ultrasonic stirring in distilled water. The resulting mixture was hydrothermally treated at 200 °C for 2 h in a Teflon-lined autoclave. The resulting powder was washed with ultrapure water to remove salts (until pH~6.5), then dried and calcined at 400 °C for 2 h in air.

The structure and morphology of both doped and non-doped TiO_2_ NPs were evaluated using a transmission electron microscope operating at 200 kV (JEOL JEM—ARM200F, JEOL Inc., Tokyo, Japan). In addition, the phase content of TiO_2_ NPs was determined by X-ray diffraction (XRD) measurements that were performed on a Bruker D8 Advance (Bruker, Hamburg, Germany), using CuK_α_ radiation (λ = 1.5406 Å). The presence of iron and nitrogen in the doped samples was evidenced through X-ray photoelectron spectroscopy (XPS) measurements carried out in an analysis chamber equipped with a 150 mm hemispherical electron energy analyzer (Phoibos, SPECS Gmbh, Berlin, Germany) and a monochromatized Al K_α1_ X-ray source (1486.74 eV). The complete procedure for the physicochemical characterization of TiO_2_ NPs was previously described [24].

In order to assess the hydrodynamic size and zeta potential, the TiO_2_ NPs suspensions were prepared at a concentration of 2 mg/mL in PBS, and of 10 and 50 µg/mL in cell culture media, sonicated for 5 min at room temperature with an ultrasonic processor UP50H (Hielscher Ultrasonics GmbH, Teltow, Germany) and then analyzed on a Malvern Zetasizer Nano-ZS instrument (Malvern Instruments, Malvern, Worcestershire, UK) by dynamic light scattering (DLS) and laser Doppler velocimetry (LDV) technologies. Three measurements have been done for each sample to determine the particles’ size and zeta potential.

### 2.2. In Vitro Cell Culture

In this experiment, normal human lung fibroblasts (MRC-5 cell line purchased from American Type Culture Collection (ATCC, Catalog Number CCL-171) were used for in vitro cytotoxicity testing. The cells were grown at 37 °C, in a humidified atmosphere with 5% CO_2_, using complete Eagle’s minimum essential medium (MEM; Gibco/Invitrogen, Carlsbad, CA, USA), containing 2 mM L-glutamine, 0.1 mM sodium pyruvate, 4.5 g/L glucose, and supplemented with 10% fetal bovine serum (FBS; Gibco/Invitrogen, Carlsbad, CA, USA). The growth medium was replaced with a fresh one every 2 days until 80% confluence was reached. Then, a 0.25% (*w*/*v*) Trypsin 0.53 mM EDTA solution (Sigma-Aldrich, St. Louis, MO, USA) was used to detach and split the cells for future sub-cultivations.

### 2.3. Cell Exposure to TiO_2_ NPs

Stock suspensions of 2 mg/mL TiO_2_ NPs were prepared by dispersing 10 mg of TiO_2_ powder in 5 mL of phosphate-buffered saline (PBS). The resulting suspensions were sonicated for 5 min at room temperature with an ultrasonic processor UP50H (Hielscher Ultrasonics GmbH, Teltow, Germany), sterilized by UV exposure for 30 min and then kept sterile until they were used. The fibroblasts were detached as described in Section 2.2, counted using a hemocytometer with a double counting chamber and an inverted phase-contrast microscope, and then seeded at 2 × 10^4^ cells/cm^2^ cell density into 24-well plates (for biocompatibility assessment), 96-well plates (for autophagy detection) or in 75 cm^2^ culture flasks (for oxidative stress analysis). The cells were allowed to adhere overnight and were then exposed to 10 and 50 µg/mL TiO_2_ NPs for 24 and 72 h. Cells grown in NP-free culture medium were used as controls for each test.

### 2.4. Cell Viability

Due to the high potential of TiO_2_ NPs to interfere with commonly used tetrazolium-based assays (e.g., MTS and MTT), the cell viability was measured using a Sulforhodamine B-based commercial kit (In Vitro Toxicology Assay Kit, Sulforhodamine B-based, St. Louis MO, USA) according to the manufacturer’s instructions. This method is used for cell density determination, based on total cellular protein content measurement. After TiO_2_ NPs exposure, the fibroblasts were fixed for one hour at 4 °C with a 50% trichloroacetic acid solution (1/4 volume of culture medium). Afterwards, cells were stained with 0.4% sulforhodamine B solution for 20 min (at room temperature), then rinsed quickly with 1% acetic acid solution. The incorporated dye was solubilized with 10 mM Tris Base solution. The absorbance was recorded at 550 nm using a microplate reader (TECAN GENios, Grödig, Austria) and the results were expressed relative to the control.

For a greater accuracy, the cell viability results were also confirmed by Trypan blue exclusion assay where 15 µL of fibroblasts suspension were mixed with 15 µL of 0.4% (*w*/*v*) Trypan blue solution, prepared in 0.81% NaCl and 0.06% (*w*/*v*) dibasic potassium phosphate (Sigma-Aldrich, St. Louis, MO, USA), and the cell viability was determined using the following formula:% Viable cells = [1.00 − (Number of blue cells/Number of total cells)] × 100(1)

The cell membrane integrity was also determined by the lactate dehydrogenase (LDH) amount released in culture medium using a commercial kit (TOX7, Sigma-Aldrich). Briefly, 100 µL of an equimolar mixture of dye, substrate and cofactor were added to 50 µL of cell culture supernatants, and then incubated for 30 min in dark. Afterwards, the reaction was stopped with 15 µL of 1 N HCl, and the absorbance was recorded at 490 nm using a microplate reader (TECAN GENios, Grödig, Austria).

Additionally, cell spreading, morphology and actin cytoskeleton dynamics were observed via fluorescence microscopy. After 24 and 72 h of treatment, the fibroblasts were fixed with 4% paraformaldehyde for 20 min and permeabilized by incubating for 1 h with a solution containing 0.1% Triton X-100 and 2% bovine serum albumin (BSA). Further, filamentous actin (F-actin) was labelled with 20 µg/mL phalloidin conjugated with FITC (Sigma-Aldrich, Munich, Germany). Images were captured using an inverted fluorescence microscope Olympus IX71 (Olympus, Tokyo, Japan).

### 2.5. Cell Lysates and Protein Concentration Assay

MRC-5 cells were harvested from flasks, washed with PBS and lysed by sonication (3 cycles × 30 s) on ice bath with an ultrasonic processor (Hielscher UP50H, Teltow, Germany). The obtained cellular homogenate was centrifuged (4 °C, 3000× *g*, 10 min) and the total protein extracts (supernatants) were collected for biochemical assays and stored at −80 °C until use.

The protein concentration of the cell lysates was determined according to the method described by Bradford [25] using the Bradford Reagent (Sigma-Aldrich, St. Louis, MO, USA) and a BSA standard curve.

### 2.6. Antioxidant Enzymes Assays

The activities of antioxidant enzymes were determined by spectrophotometric methods using SPECORD 200 PLUS double-beam spectrophotometer from AnalytikJena. Catalase (CAT) (EC 1.11.1.6) activity was measured by monitoring the decrease in absorbance of H_2_O_2_ at 240 nm, as described in Aebi’s method [26]. One unit of CAT activity represented the amount of enzyme that catalyzed the conversion of 1 µmole of H_2_O_2_ in 1 min under standard conditions. Glutathione peroxidase (GPx) (EC 1.11.1.9) activity was assessed according to the method of Beutler [27] by monitoring at 340 nm a coupled reaction with glutathione reductase that catalyzed NADPH oxidation. One unit of GPx activity was defined as the quantity of enzyme that catalyzes the transformation of 1 µmole of NADPH per minute. Glutathione S-transferase (GST) (EC 2.5.1.18) activity was determined as described in the method of Habig et al. [28] by measuring the rate of 1-chloro-2,4-dinitrobenzene (CDNB) conjugation with reduced glutathione (GSH) at 340 nm. One unit of GST activity was defined as the amount of enzyme that produced 1 µmole of conjugated product per minute. All data were obtained in physiological conditions at pH 7.4 and room temperature (25 °C), and the final results were calculated as specific enzymatic activities (units/mg of protein) and expressed relative to the control.

### 2.7. Reduced Glutathione Assay

The intracellular GSH concentration was determined using a Glutathione Assay kit (Sigma-Aldrich, St. Louis, MO, USA). In the first step, the proteins from cell lysates were precipitated with 5% sulfosalicylic acid (Sigma-Aldrich, St. Louis, MO, USA) and removed by centrifugation (4 °C, 10,000× *g*, 10 min). Further, the samples were incubated with 5,5′-dithiobis-2-nitrobenzoic acid (DTNB) at room temperature for 5 min to allow the reduction of DTNB into 5-thio-2-nitrobenzoic acid (TNB). The optical density was read at 405 nm using a microplate reader (TECAN GENios, Grödic, Germany) and the GSH content was expressed as nmoles/mg protein and represented relative to the control.

### 2.8. Lipid Peroxidation

Malondialdehyde (MDA) is one of the most widely used markers of lipid peroxidation. MDA level was measured using the fluorimetric method described previously by our group [29]. Volumes of 200 µL of cell lysates were mixed with 700 µL of 0.1 N HCl and incubated at room temperature for 20 min. Further, 900 µL of 0.025 M thiobarbituric acid were added and the mixture was incubated at 37 °C for 65 min. The relative fluorescence units were recorded using excitation wavelength set at 520 nm and emission wavelength set at 549 nm (FP-750 Spectrofluorometer, Jasco, Tokyo, Japan) and converted to nmoles of MDA using a standard curve of 1,1,3,3-tetramethoxypropane. Subsequently, the final MDA level was expressed as nmoles of MDA/mg protein and all results were represented relative to the control.

### 2.9. Intracellular Reactive Oxygen Species Level

The intracellular reactive oxygen species (ROS) level was determined using a fluorescent compound 2′,7′-dichlorofluorescein diacetate (DCFH-DA, Sigma-Aldrich, St. Louis, MO, USA). After TiO_2_ NPs exposure, the cells were washed with PBS and incubated with the dye for 30 min at 37 °C. Afterwards, the excess dye was removed, and the fibroblasts were resuspended in PBS and detached by scraping. The fluorescence was quantified using a fluorimeter (excitation wavelength = 488 nm and emission wavelength = 515 nm, FP-750 Spectrofluorometer, Jasco, Tokyo, Japan). All results were expressed relative to the control after fluorescence intensity was reported to the number of viable cells of each sample.

### 2.10. Autophagy Detection

Induction of autophagy in cells exposed to TiO_2_ NPs was established using the Autophagy/Cytotoxicity Dual Staining Kit (Cayman Chemical, Ann Arbor, MI, USA) that allowed the detection of autophagic vacuoles in cultured cells, but also the detection of dead cells. This technique uses the monodansylcadaverine (MDC) fluorescent compound to label autophagic vacuoles and propidium iodide (PI) as a marker for cell death. After 24 and 72 h of exposure, the culture medium was aspirated and the cells were incubated with PI for 2 min at room temperature, and then with MDC solution for 10 min at 37 °C. In parallel, MRC-5 cells previously incubated with 20 μM tamoxifen, a well-known inducer of autophagy, were also labeled. Further, the MDC solution was replaced with wash buffer and cells were immediately visualized by an Olympus IX71 fluorescence microscope (triple filter TRITC/FITC/DAPI, Olympus, Tokyo, Japan). The dead cells were revealed in red, while the autophagic vacuoles were shown in silver. The fluorescence was quantified using the image processing and analysis software ImageJ 1.48 (National Institutes of Health, Bethesda, MD, USA). The results were expressed as changes in fluorescence intensity compared to the control, analyzing 50 cell fields per sample.

### 2.11. Statistical Analysis

Data were expressed as mean value ± standard deviation (SD) of three independent experiments. Statistical differences between the samples and control were evaluated by Student’s *t*-test using the GraphPad Prism software (version 5; GraphPad Software, Inc., La Jolla, CA, USA), and a value of *p* < 0.05 was expressed as being statistically significant.

## 3. Results

### 3.1. Physicochemical Features of TiO_2_ NPs

The crystal phase structure of the TiO_2_ NPs used in the present study was investigated using XRD. The X-ray diffractograms presented in Figure 1B, and the corresponding Rietveld refinement results summarized in Table 1, showed that both P25 and P25-Fe(1%)–N NPs consist of two phases: anatase (~83%) and rutile (~17%). This result is consistent with the well-known biphasic structure of commercial TiO_2_ P25. Additionally, the crystallite size of anatase phase (30 nm) was generally lower compared to the rutile one (50 nm).

The presence of iron and nitrogen atoms in the P25-Fe(1%)–N NPs was revealed through the XPS measurements. The characteristic spectra of Ti 2p, O 1s, Fe 2p and N 1s, with their corresponding binding energies, are represented in Figure 1B and Table 2, respectively. The main components of Ti 2p (458.65 eV) and O 1s (529.98 eV) in the pure TiO_2_ P25 NPs revealed a stoichiometric TiO_2_ with a O/Ti ratio of 2.02. Introducing the doping atoms into the structure of TiO_2_ led to a shift towards lower binding energies for both Ti 2p and O 1s, due to an oxygen deficiency as a consequence of the nitrogen doping. Normally, the Fe 2p signal is ~710 eV, very similar to the value reported in the present study, assigning this to the presence of Fe^3+^ in the synthesized P25-Fe(1%)–N NPs. Additionally, an extra component present in the Ti 2p spectrum of P25-Fe(1%)–N sample at ~460.37 eV can be associated with the formation of a Fe–O–Ti bond. A more detailed analysis of these two types of TiO_2_ NPs (pure and Fe–N co-doped P25 Degussa) was presented in our previous work [24,30].

The primary size and shape of the TiO_2_ NPs were determined by electron microscopy imaging. The powder morphology of P25-based nanophotocatalysts as well as the corresponding particle distribution are presented in Figure 2. As shown by transmission electron microscopy (TEM) images, the P25 particles have a polyhedral shape with round corners, while the mean particle size was 29 ± 15 nm for pure P25 NPs and 30 ± 10 nm for P25-Fe(1%)–N NPs, these results being in good agreement with the XRD results presented in Table 1.

### 3.2. Agglomeration State and Colloidal Stability of TiO_2_ NPs

The Z-average represents the intensity weighted mean hydrodynamic size of the ensemble collection of particles measured by DLS, being a reliable measure of the average size of a particle size distribution. Our measurements revealed that both types of NPs have a great tendency to aggregate in PBS, the values being higher than 2 µm (Table 3). In culture medium, the NPs have lower diameters compared to those in PBS, proving that the serum proteins were able to induce a higher dispersity of particles by steric repulsion forces. Additionally, the P25 NPs had higher values compared to doped NPs for all times of incubation, suggesting that co-doping procedure Fe and N was able to provide better dispersed particles. By increasing the time of incubation, the Z-average values elevated for all types of particles and concentrations tested. However, compared to the time-dependent increase in hydrodynamic size, there was no concentration-dependent increase in the diameter.

Regarding the zeta potential, there was an important difference between the medium used for dispersion. The P25 NPs and P25-Fe(1%)–N NPs dispersed in PBS registered a value of approx. −25 mV and −22 mV, respectively, suggesting their strong negative surface charge. By incubating NPs in culture medium supplemented with animal serum, the zeta potential increased to ~−9.5 mV, regardless of the NP type or concentration tested. This change confirmed the interaction between NPs and serum proteins, which altered the surface charge. Increasing the time of incubation in this type of medium, the zeta potential was not changed very much, suggesting that the interaction with proteins was very fast and irreversible.

### 3.3. Influence of TiO_2_ NPs on Cell Viability of Human Lung Fibroblasts

The influence of TiO_2_ NPs on the viability and membrane integrity on normal human lung fibroblasts was evaluated by in vitro methods. For this purpose, MRC-5 cells were exposed to two different concentrations (10 and 50 µg/mL) of commercially available TiO_2_ P25 Degussa NPs and Fe(1%)-N doped TiO_2_ P25 NPs for 24 and 72 h. These concentrations were selected based on key studies from available literature [31,32]. Additionally, it is worth noting that due to the predisposition of TiO_2_ NPs to form large aggregates (>100 nm) in suspension, higher doses may show no toxic effects.

Figure 3A revealed that no significant changes in cell viability were observed after 24 h of exposure to TiO_2_ NPs compared to the control (untreated cells). In contrast, after 72 h, both doses of P25-Fe(1%)-N NPs decreased cell viability by almost 15% compared to the control and even compared to cells exposed to 10 µg/mL commercial TiO_2_ P25 NPs. The presence of 50 µg/mL TiO_2_ P25 NPs in the cell culture media also induced a 10% decrease in cell viability of MRC-5 fibroblasts.

To better describe the cytotoxicity of TiO_2_ NPs, the LDH leakage level was measured. Figure 3B shows the amount of LDH released into the culture medium by fibroblasts exposed to nanophotocatalysts. In contrast to the results obtained by the Sulforhodamine B-based assay, the LDH levels suggested that TiO_2_ NPs did not affect the cell membrane integrity of MRC-5 cells in the first 24 h of exposure (Figure 3). Instead, after 72 h, both non-doped TiO_2_ P25 and Fe-N-containing TiO_2_ NPs lead to a slight dose-dependent reduction in the amount of LDH released in the culture medium compared to the control, which could be correlated with a proportional decline in cell number compared to the control. However, these two analyses showed that non-doped TiO_2_ P25 NPs had no effect on the viability of MRC-5 lung fibroblasts. Additionally, the results suggest that P25-Fe(1%)-N NPs exerted a more pronounced toxicity on cell viability, but their effect can be considered only moderate.

Actin cytoskeleton analysis can provide important information about the influence of TiO_2_ NPs on cell morphology. Morphological changes of MRC-5 cells after exposure to TiO_2_ P25 and P25-Fe(1%)-N NPs were evidenced by fluorescence microscopy. Figure 3C,D show the actin cytoskeleton organization, which are consistent with the results of the cell viability tests shown in Figure 3A,B. Thus, it was observed that human lung fibroblasts maintained their specific elongated morphology after 24 and 72 h of TiO_2_ NPs exposure with no significant differences between the samples and control or between the types of nanophotocatalysts. Moreover, the presence of TiO_2_ NPs in the culture medium did not change the cellular behavior.

### 3.4. ROS Generation by TiO_2_ NPs and Cellular Antioxidant Defense

In order to assess if TiO_2_ P25 and P25-Fe(1%)-N NPs can induce oxidative stress, the level of intracellular ROS was measured using the fluorescence intensity of dichlorofluorescein (DCF). The results presented in Figure 4A showed that TiO_2_ P25 NPs had a significant effect of stimulating ROS production, their level increasing up to five times compared to the control after 72 h of exposure to the higher dose of 50 µg/mL. This major difference between the two samples suggests that Fe-doped TiO_2_ NPs can be considered harmless compared to non-impregnated ones (P25 Degussa).

ROS generated as a result of the cellular metabolism, as well as due to nanomaterials exposure, can be counteracted by non-enzymatic and enzymatic antioxidant systems. In order to evaluate the non-enzymatic antioxidant defense system responsible for the cell protection against oxidative injuries induced by ROS, the GSH levels were measured. The 24 h exposure of MRC-5 fibroblasts to photocatalytic NPs did not result in major changes in GSH levels (Figure 4B), but after 72 h of incubation with 10 µg/mL P25-Fe(1%)-N an increase in GSH level by almost 40% over the control was registered. In contrast, the higher dose of the same nanophotocatalyst (50 µg/mL) induced a GSH depletion. It was also observed that GSH concentration was similar to the control in the case of lung cells exposed to TiO_2_ P25 NPs for 24 and 72 h.

The cellular defense against oxidative damage was also assessed by measuring the enzymatic activities of CAT, GST and GPx. As it can be seen in Figure 5A, the CAT activity recorded a slight concentration-dependent increase (up to 10%) after 24 h of incubation of MRC-5 cells with TiO_2_ P25 NPs. In contrast, P25-Fe(1%)-N NPs induced different changes depending on the tested concentration. In the first 24 h, the presence of a lower concentration (10 µg/mL) of Fe/N-doped TiO_2_ NPs led to a slight 6% increase in CAT activity compared to the control, while the higher dose resulted in a decrease in enzymatic activity by almost 10%. After incubation of human lung fibroblasts for 72 h in the presence of TiO_2_ NPs, a decrease in CAT activity was observed both in cells exposed to TiO_2_ P25 NPs and P25-Fe(1%)-N NPs, but more pronounced in cells exposed to 10 µg/mL of P25-Fe(1%)-N NPs.

Regarding the activity of glutathione-dependent enzymes the same trend was observed. Specifically, the GPx activity shown in Figure 5B registered an insignificant increase (≤20%) for all samples compared to the control after the 24 h exposure interval. Instead, after 72 h, there was a diminution of GPx level up to 30% compared to the control.

Additionally, Figure 5C revealed that the GST activity did not change significantly in the first 24 h of exposure to TiO_2_ NPs, registering only a slight increase compared to the control (≤15%), while after 72 h a slight decrease in GST activity was observed. All these changes suggest a slight stimulation and activation of the antioxidant defense mechanism by both types of TiO_2_ NPs.

### 3.5. Effect of TiO_2_ NPs on Lipid Peroxidation of Cellular Membranes

The effects of TiO_2_ NPs on lipid peroxidation, especially of cellular membranes of lung fibroblasts, are shown in Figure 6. The first 24 h in the presence of the nanophotocatalysts did not induce significant changes of MDA level in MRC-5 cells, but after 72 h of incubation, an elevation of MDA content by ~30% compared to the control was registered only for the higher dose (50 µg/mL) of both TiO_2_ P25 NPs and P25-Fe(1%)-N NPs. These observations showed that the MDA level increased in a time- and dose-manner, but no major differences were observed between the two types of nanophotocatalysts.

### 3.6. Autophagic Effect of TiO_2_ NPs on MRC-5 Cells

Figure 7 revealed that the fluorescence intensity of MDC staining in MRC-5 cells after exposure to TiO_2_ NPs increased in a dose- and time-dependent manner. Regarding the type of nanoparticles tested, significant differences were observed between pure P25 Degussa NPs and those doped with Fe and N atoms. Microscopy images showed that both doses of P25 NPs induced the activation of autophagy in MRC-5 lung cells even after the first 24 h of exposure, demonstrating once again that Fe-N doping could reduce their cytotoxic effect on human lung fibroblasts.

## 4. Discussion

The exponential growth of nanotechnology has led to numerous studies focusing on the risks associated with the NPs’ use. However, despite our growing exposure to different NP types, there is so much less information on nanomaterials’ safety compared to the research on nanoscaled particle applications.

The toxicity of NPs depends on many factors, including size, shape, surface area, surface charge as well as aggregation state, determined mainly by the specific parameters of each synthesis method [33]. These physicochemical properties have been shown to affect the distribution and accumulation of NPs in different tissues and organs, at the same time altering their molecular interactions with various macromolecules [34]. Therefore, there are many methods to prevent or limit the toxic effects of NPs. For example, modifying the shape, size and surface of particles can lead to NPs with the desired properties, but without toxic effects [35].

Due to the high interest generated by the photocatalytic applications of TiO_2_ NPs in different fields such as medicine, food industry or cosmetics, we comparatively studied the toxicity of pure TiO_2_ P25 NPs and the same NPs co-doped with Fe (1%) and N atoms in order to increase the photocatalytic activity in the visible light spectrum. The synergistic effect of Fe-N doping was explained by the fact that nitrogen atoms narrow of the band gap of TiO_2_, which allows absorption at longer wavelengths in visible light, while Fe helps to increase the efficiency of charge separation [36].

From the obtained results, it could be observed that neither TiO_2_ P25 Degussa NPs nor Fe-N-doped TiO_2_ NPs have significantly influenced the viability of MRC-5 lung fibroblasts. However, in some cases there was a very small decrease in LDH level after 72 h of exposure to nanophotocatalysts. It is known that liquid-suspended TiO_2_ NPs can adsorb soluble proteins such as albumin [37,38] or γ-globulin [39]. LDH is a cytosolic enzyme released in the culture medium from cells following damage to their cell membrane. Zaqout et al. showed that TiO_2_ NPs can interact with LDH in the Dulbecco’s Modified Eagle Medium (DMEM) culture medium [40], which could explain the lower level of the enzyme in the supernatant from NP-treated MRC-5 cells. The influence of several types of NPs, including TiO_2_, on the results of the LDH test was also demonstrated by the study of Holder et al. using 16HBE14o cells [41]. However, it is not clear whether this interaction with NPs affects the enzymatic activity of LDH. Additionally, the small discrepancy between the results of the two techniques (LDH and Sulforhodamine B-based assays) as well as the more pronounced toxic effects observed in the case of lower NP concentrations in some analyses could be explained by the aggregation and/or agglomeration phenomena that occur in TiO_2_ NPs colloidal solutions. Due to them, the interactions with cells are less harmful. Smaller and single NPs usually have a more pronounced toxic effect [42,43], but there are also exceptions to this rule [44].

The way by which NPs can enter into cells is a key factor in determining their toxicity. Endocytosis dependent cellular uptake of NPs can cause much more damage to the cell compared to NPs that get into the cell through a different route such as passive transport [45]. Several studies confirm the major role of actin cytoskeleton in the mechanism of endocytosis [46,47,48]. Thus, the reorganization of the actin cytoskeleton can be considered an indicator of the internalization of NPs. In this study, the actin cytoskeleton dynamic changes were analyzed by fluorescence microscopy in MRC-5 cells exposed to TiO_2_ NPs, with the most representative images being shown in Figure 3. No major differences were observed between the control and the two samples. In all experimental variants, the organization of the actin cytoskeleton was similar to that observed in the control, the actin filaments being organized in dense bundles. The images also suggested that the specific morphology of MRC-5 lung fibroblasts was not affected following exposure to TiO_2_ NPs. A similar result was observed after exposure of HaCaT keratinocytes to the same TiO_2_ concentrations used in this experiment (10 and 50 μg/mL) [49]. Moreover, Huerta-Garcia et al. showed that glial cells uptake TiO_2_ NPs from the culture medium through pseudopods and vesicles whose formation is dependent on the rapid and reversible reorganization of the actin cytoskeleton [50]. In the case of A549 epithelial lung cells, the internalization of TiO_2_ NPs had a significant effect, the architecture of the actin and tubulin cytoskeleton being severely and irreversible disturbed [51]. Taking in consideration all these results, it could be possible that P25 and P25–Fe(1%)-N NPs have not been internalized in MRC 5 fibroblasts.

In the case of metal oxide NPs, the mechanism of their toxicity is closely related to the release of metal ions in contact with cells or biological fluids [52]. In addition, the photocatalytic capacity is attributed to the generation of active oxygen species on the surface of photoexcited TiO_2_ [53]. Although the fundamental process of ROS production during photoexcitation involving photo-induced electrons and holes was first discovered in the early 20th century by Goodeve and Kitchener [54], the mechanism for intracellular ROS generation by NPs is still unclear. Our results showed that the ROS formation in cells exposed to TiO_2_ NPs was time- and dose-dependent, but a major difference could be observed between the two samples. The highest dose (50 µg/mL) of TiO_2_ P25 NPs had a significant stimulating effect on ROS production compared to P25-Fe(1%)-N photocatalyst. The toxicity of P25 Degussa has been reported previously and this may be due to the mixture of anatase and rutile phase in a 3:1 ratio, both phases exhibiting a synergistic effect on the UV photocatalytic activity of the TiO_2_ NPs [55,56]. In contrast, our present and previous studies [24] showed that P25-Fe(1%)-N NPs have a superior photocatalytic performance (both in UV and visible light) and reduced cytotoxicity. The fact that Fe-doping inhibits ROS generation has also been observed by Ghiazza et al. that performed experiments on human HaCaT keratinocytes [57]. The type of metal used for doping is very important to obtain the cytotoxicity-reducing effect. Ag-doped TiO_2_ NPs have been shown to increase intracellular ROS levels in HepG2, A549 and MCF-7 tumor cells, and the effect is increasingly pronounced with the amount of Ag. However, the same study showed that Ag-doped TiO_2_ NPs did not induce an increase in ROS levels in normal cells such as IMR-90 fibroblasts or primary rat hepatocytes [58].

TiO_2_ NPs may alter ROS production and thereby may cause interference in biological antioxidant defense responses. Normally, the superoxide radicals are converted into hydrogen peroxide (H_2_O_2_) under the defensive action of the antioxidant enzyme called superoxide dismutase (SOD), and then converted to water in the presence of other enzymes, CAT and GPx. However, if the number of radicals is too high in relation to cell antioxidant capacity, part of these ROS does not decompose, but enters the nucleus, causing oxidative damage of biomolecules and, subsequently, cell death [59]. In our study, oxidative stress was initiated by P25 and P25-Fe(1%)-N in a time- and dose-dependent manner, but the decrease in GSH levels only after 72 h of exposure to 50 µg/mL of Fe-N-doped P25 NPs suggests that photocatalysts were well tolerated. The lack of significant changes in catalase in the first 24 h and the decrease in its activity after 72 h of exposure to NPs also suggest that large amounts of H_2_O_2_ were not formed. Additionally, there was an increase in GPx and GST levels after 24 h of incubation with the TiO_2_ NPs, which probably resulted from an oxidative attack initiated by the reduction in the cellular antioxidant defense mechanisms. This increase in the activity of the two antioxidant enzymes is also well correlated with the increase in the level of lipid peroxidation in cells exposed to TiO_2_ NPs, indicating that they are probably involved in MDA and H_2_O_2_ detoxification [60].

An interesting observation regarding the hydrodynamic size of these NPs was that there was no correlation between the concentration tested and the Z-average of P25 NPs (Table 3). Additionally, higher values were measured after 72 h of incubation for the 10 µg/mL concentration compared to 50 µg/mL of P25-Fe(1%)–N NPs. This suggests the generation of higher aggregates in the culture medium at this time interval, and it could explain the lower activities of CAT and GPx after 72 h of incubation with 10 µg/mL compared to the values obtained for the 50 µg/mL concentration.

On the other hand, our results showed that autophagy was well correlated with increased ROS levels in MRC-5 cells treated with TiO_2_ NPs, especially with pure P25 Degussa. It has been shown that the connection of ROS and autophagy and the balance between them is an important indicator for cellular homeostasis in such experiments involving cells exposed to NPs [61]. ROS production appears to be the upstream modulator of autophagy and several studies revealed that the inhibition of ROS with N-acetyl-L-cysteine caused defective autophagy induced by NPs in HEK293 [62], HepG2 and human glioma cells [63,64]. It should be noted that depending on the concentration used, NPs can induce different autophagic effects. Usually, a low dose of NPs activates mTOR-mediated autophagy that plays a protective antioxidative role, whereas high dose could cause autophagic cell death [65]. However, NP-induced autophagy is rarely associated with cell survival. In our study, we showed that this programmed cell death pathway was primarily activated in lung fibroblasts exposed to P25 Degussa, demonstrating once again the efficiency of Fe-N doping in reducing their cytotoxicity.

## 5. Conclusions

Our results showed that the Fe-N-doping of TiO_2_ P25 NPs can reduce their toxic effect on human lung fibroblasts. The oxidative stress was initiated in a time- and dose-dependent manner with major differences between pure P25 and P25-Fe(1%)-N NPs. Additionally, a decreased rate of autophagic vacuoles formation was observed in cells exposed to Fe-N-doped P25 NPs compared to the pure ones, providing information that may contribute as a reference for future studies on photocatalytic applications. In conclusion, our research contributes to a better understanding of the mechanisms of interaction between functionalized TiO_2_ P25 NPs and in vitro biological systems and can also help to obtain more biocompatible and efficient TiO_2_-based nanomaterials.

## Figures and Tables

**Figure 1 nanomaterials-12-00770-f001:**
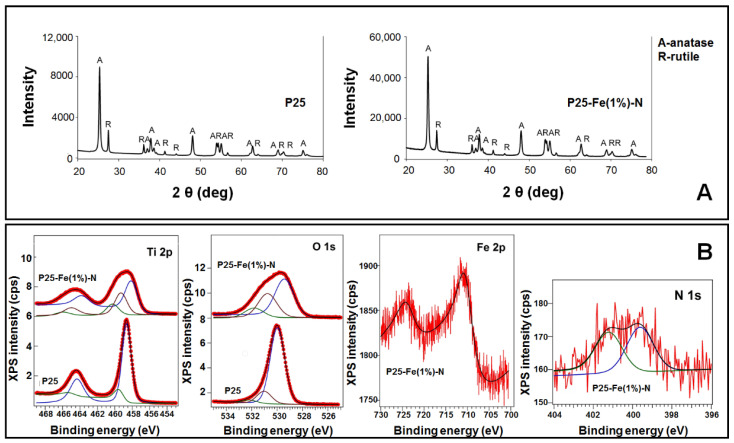
Physicochemical characteristics of TiO_2_ NPs. (**A**) X-ray diffraction (XRD) patterns and phase assignment. (**B**) The X-ray photoelectron spectroscopy (XPS) spectra of the P25 and P25–Fe(1%)-N NPs samples.

**Figure 2 nanomaterials-12-00770-f002:**
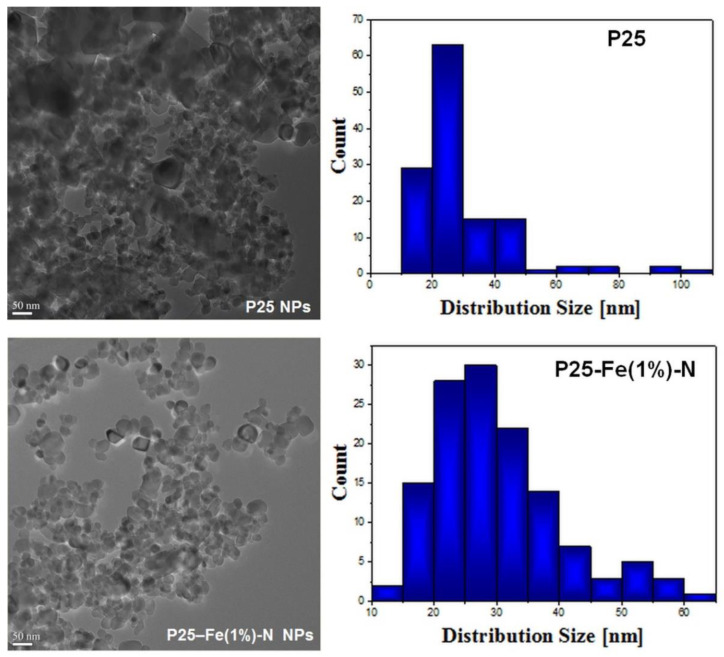
Transmission electron microscopy (TEM) images and particle distribution diagrams of the P25 and P25–Fe(1%)-N NPs samples.

**Figure 3 nanomaterials-12-00770-f003:**
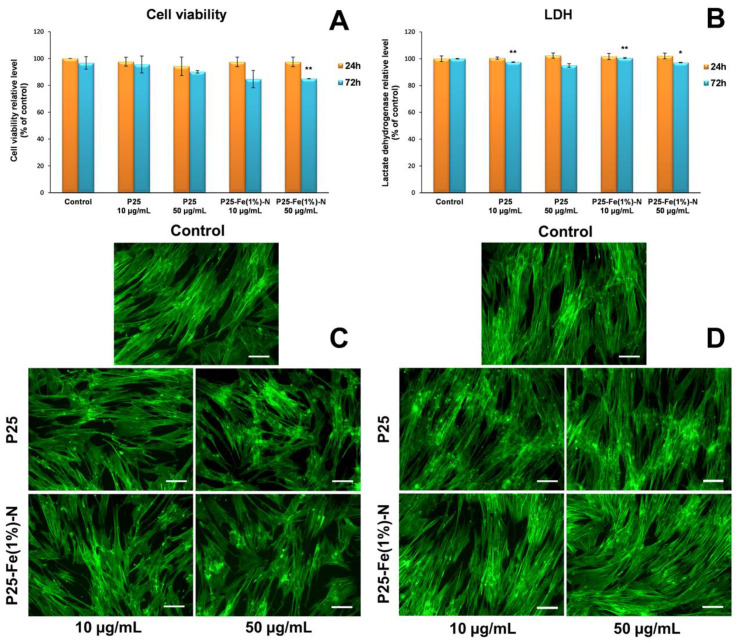
Biocompatibility of different concentrations (10 and 50 μg/mL) of the two TiO_2_ NPs samples: P25 Degussa and P25-Fe(1%)-N, as shown by cell viability (**A**) and LDH release (**B**) after 24 and 72 h of exposure on normal lung fibroblasts. Results are expressed as means ± standard deviation (SD) (*n* = 3) and represented relative to the untreated cells (control). * *p* < 0.05 and ** *p* < 0.01 compared to control. Actin cytoskeleton organization of lung fibroblasts after 24 (**C**) and 72 h (**D**) of incubation with different concentrations (10 and 50 μg/mL) of the two TiO_2_ NPs samples: P25 Degussa and P25-Fe(1%)-N. F-actin (green) was labeled with phalloidin-fluorescein isothiocyanate (FITC). Scale bar: 100 μm.

**Figure 4 nanomaterials-12-00770-f004:**
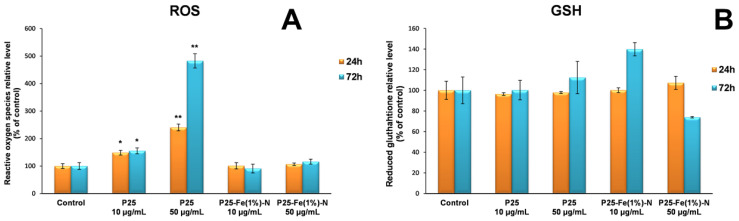
ROS (**A**) and GSH (**B**) levels in normal lung fibroblasts after 24 and 72 h of exposure to different concentrations (10 and 50 μg/mL) of the two TiO_2_ NPs samples: P25 Degussa and P25-Fe(1%)-N. Results are expressed as means ± standard deviation (SD) (*n* = 3) and represented relative to the untreated cells (control). * *p* < 0.05 and ** *p* < 0.01 compared to control.

**Figure 5 nanomaterials-12-00770-f005:**
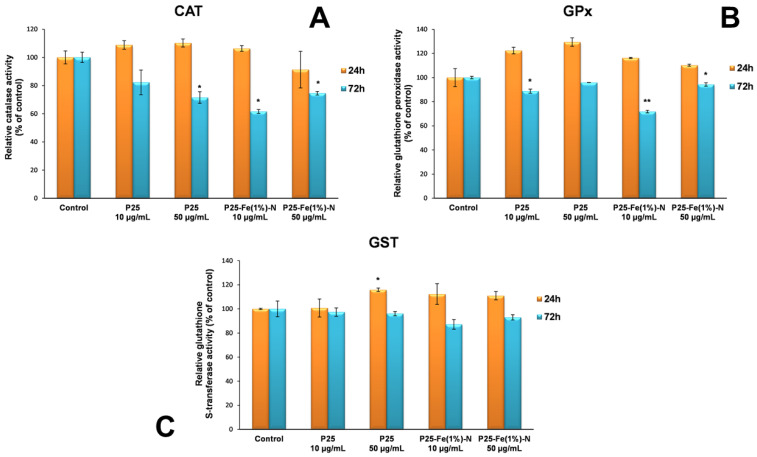
Relative levels of catalase (**A**) and glutathione-dependent enzymes (glutathione peroxidase and glutathione S-transferase) (**B**,**C**) specific activities in normal lung fibroblasts exposed to different concentrations (10 and 50 μg/mL) of the two TiO_2_ NPs samples: P25 Degussa and P25-Fe(1%)-N for 24 h and 72 h. Results are expressed as means ± standard deviation (SD) (*n* = 3) and represented relative to the untreated cells (control). * *p* < 0.05 and ** *p* < 0.01 compared to control.

**Figure 6 nanomaterials-12-00770-f006:**
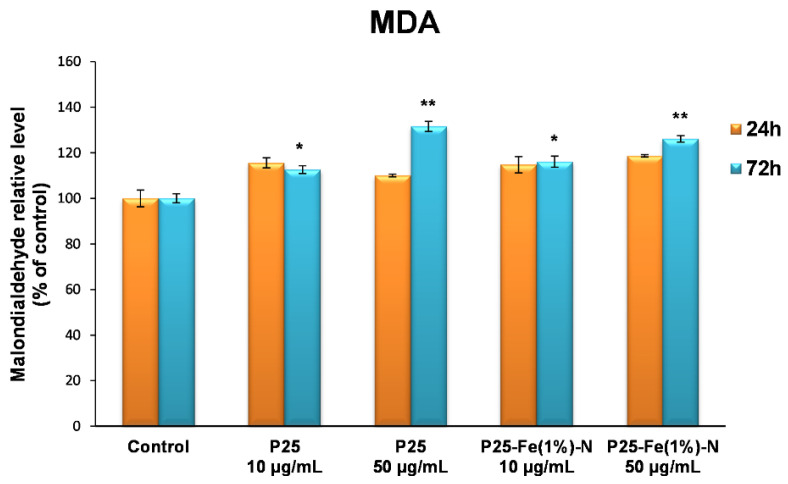
Malondialdehyde (MDA) levels in normal lung fibroblasts exposed to different concentrations (10 and 50 μg/mL) of the two TiO_2_ NPs samples: P25 Degussa and P25-Fe(1%)-N for 24 h and 72 h. Results are expressed as means ± standard deviation (SD) (*n* = 3) and represented relative to the untreated cells (control). * *p* < 0.05 and ** *p* < 0.01 compared to control.

**Figure 7 nanomaterials-12-00770-f007:**
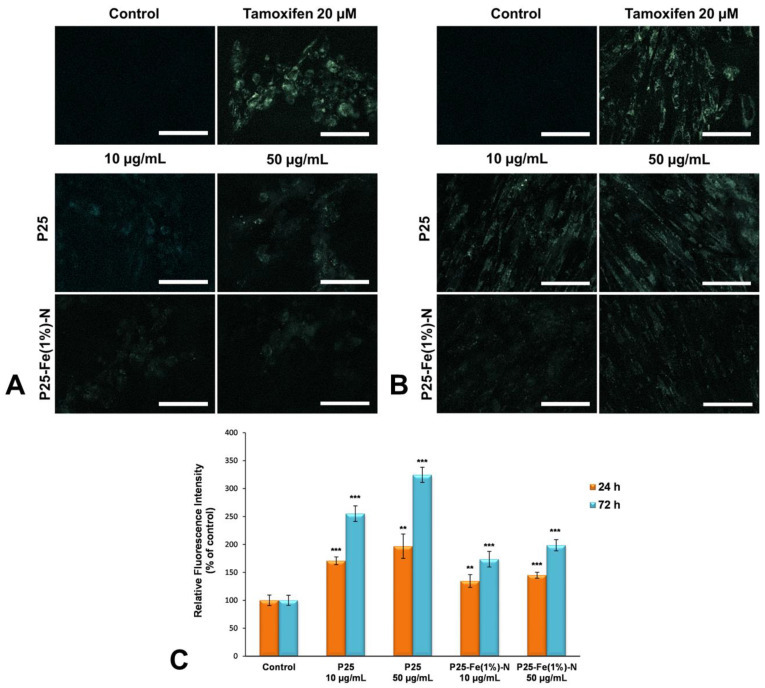
Autophagy induced in MRC-5 cells by TiO_2_ NPs. Representative images of fluorescence microscopy showing the autophagic vacuoles induced in normal lung fibroblasts after exposure to different concentrations (10 and 50 μg/mL) of the two TiO_2_ NPs samples: P25 Degussa and P25-Fe(1%)-N for 24 (**A**) and 72 h (**B**). Scale bar: 100 µm; (**C**) Quantification of MDC fluorescence intensity by Image J software (National Institutes of Health, Bethesda, MD, USA); Results are expressed as means ± standard deviation (SD) (*n* = 3) and represented relative to the untreated cells (control). ** *p* < 0.01 and *** *p* < 0.001 compared to control.

**Table 1 nanomaterials-12-00770-t001:** Lattice parameters, crystallite size, phase assignment and relative abundance of the P25 and P25–Fe(1%)-N NPs samples.

Sample	Lattice Parameters (Å)			Crystallite Size (nm)	Phase Assignment/ Abundance (wt%)
	*a*	*b*	*c*		
P25	3.7891	-	9.5165	32.8	Anatase (83.1)
	4.5977	-	2.9598	71.0	Rutile (16.9)
P25-Fe(1%)-N	3.7876	-	9.5139	28.6	Anatase (83.0)
	4.5942	-	2.9632	36.4	Rutile (17.0)
Errors	±0.0005	-	±0.0005	±1.5	±1.4

**Table 2 nanomaterials-12-00770-t002:** The binding energy values extracted from the deconvolutions of the X-ray photoelectron spectroscopy (XPS) spectra of the P25 and P25–Fe(1%)-N NPs samples.

Sample	Ti 2p_3/2_	O 1s	Fe 2p_3/2_	N 1s
	Binding Energy (eV)			
P25	458.65	529.98	-	-
	459.63	531.03	-	-
	-	532.16	-	-
P25-Fe(1%)-N	458.08	529.42		399.62
	459.30	530.75	710.40	401.19
	460.27	531.81	-	-

**Table 3 nanomaterials-12-00770-t003:** Hydrodynamic diameter and zeta potential values measured for TiO_2_ NPs dispersed in PBS or culture medium (MEM + 10% FBS) for different periods of time. Results are represented as means ± standard deviation (SD) (*n* = 3).

Condition	Samples	Z-Average (d.nm)	Zeta Potential (mV)
5 min PBS	P25 NPs 10 µg/mL P25 NPs 50 µg/mL P25-Fe(1%)–N NPs 10 µg/mL P25-Fe(1%)–N NPs 50 µg/mL	3327 ± 319.6 2539 ± 387.5 3582 ± 654.8 2333 ± 139.0	−23.60 ± 0.42 −25.30 ± 1.20 −22.00 ± 2.05 −22.20 ± 2.40
5 min MEM + 10% FBS	P25 NPs 10 µg/mL P25 NPs 50 µg/mL P25-Fe(1%)–N NPs 10 µg/mL P25-Fe(1%)–N NPs 50 µg/mL	683 ± 6.5 1490 ± 53.7 261 ± 204.8 766 ± 183.0	−9.55 ± 0.31 −9.55 ± 0.62 −9.46 ± 0.07 −9.64 ± 0.50
24 h MEM + 10% FBS	P25 NPs 10 µg/mL P25 NPs 50 µg/mL P25-Fe(1%)–N NPs 10 µg/mL P25-Fe(1%)–N NPs 50 µg/mL	2416 ± 1199.0 1479 ± 14.8 677 ± 253.0 1006 ± 108.3	−9.38 ± 1.02 −10.40 ± 1.53 −9.85 ± 1.63 −10.20 ± 0.00
72 h MEM + 10% FBS	P25 NPs 10 µg/mL P25 NPs 50 µg/mL P25-Fe(1%)–N NPs 10 µg/mL P25-Fe(1%)–N NPs 50 µg/mL	3971 ± 360.6 2445 ± 451.6 2978 ± 1063.0 1938 ± 14.8	−11.60 ± 0.85 −11.40 ± 0.63 −8.14 ± 0.94 −10.70 ± 1.46

## Data Availability

Data are available on request from the corresponding author.

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
