# Peer review of "Could Iron-Nitrogen Doping Modulate the Cytotoxicity of TiO2 Nanoparticles?"

_nanomaterials, 2022, doi:10.3390/nano12050770_

Round 1

Reviewer 1 Report

The principal objective of this study was to evaluate the iron-nitrogen doping-based modulation of the cytotoxicity of TiO2 nanoparticles. It is a well-conducted study with valuable results in in vitro; however, there are some concerns;

  1. It needs to be carefully checked for language and grammar.
  2. A graphical abstract could be helpful for better illustration of study procedure.
  3. It’s a good idea to evaluate the effects of these NPs on DNA strands through comet assay or other geno-toxicity tests.
  4. Maybe evaluation of autophagic markers as well as Beclin 1 through gene expression analysis could be helpful in interpretation of authophagic induction effects of NPs.
  5. Why was not the study performed on an animal model? Or alternatively, 3D-culture instead of monolayer cells that is more similar to in vivo models?

6-      In the Introduction, the authors need to elaborate on the role of oxide nanoparticles as theranostic tools in the fight against different diseases by citing and briefly discussing the following papers (DOI: 10.1016/j.molliq.2018.05.105, DOI: 10.1007/s00339-021-04759-4, DOI: 10.1016/j.molstruc.2018.04.016).

Author Response

The principal objective of this study was to evaluate the iron-nitrogen doping-based modulation of the cytotoxicity of TiO2 nanoparticles. It is a well-conducted study with valuable results in in vitro; however, there are some concerns;

1. It needs to be carefully checked for language and grammar.

Response: Thank you very much for your careful revision of this article. The English language spell check has been performed throughout all manuscript.

2. A graphical abstract could be helpful for better illustration of study procedure.

Response: Thank you very much for your suggestion. A graphical abstract was added to the revised manuscript.

3. It’s a good idea to evaluate the effects of these NPs on DNA strands through comet assay or other geno-toxicity tests.

Response: Thank you very much for your suggestion. We have already considered the study of the genotoxic potential of these types of nanoparticles for the next article.

4. Maybe evaluation of autophagic markers as well as Beclin 1 through gene expression analysis could be helpful in interpretation of authophagic induction effects of NPs.

Response: Thank you very much for your competent remark. In our previous studies, we tried to evaluate the effect of TiO2 NPs on the expression of Beclin-1 protein by Western Blot analysis, but the experiment failed. This may be due to the low concentration of this protein in MRC-5 cells. For future studies we will consider other proteins involved in autophagic induction such as ATGs and LC3.

5. Why was not the study performed on an animal model? Or alternatively, 3D-culture instead of monolayer cells that is more similar to in vivo models?

Response: In the last decade there has been an exponential increase in the scientific focus given towards enhancing the physiological relevance of in vitro systems and expanding their applications to support the movement towards replacing, reducing, and refining (3Rs) the use of in vivo experiments. The reason why we did not use 3D-culture instead of monolayer cells is that 2D cell cultures are cheaper and easier to observe and analyze than some 3D cell culture systems. In addition, 3D-cultures are more suitable for tissue- and organ-level mechanisms, cell-cell interactions or trafficking across different tissue barriers. In vitro 2D cell cultures are common and widely used for testing the toxicity of chemical drugs and nanomaterials.

6-      In the Introduction, the authors need to elaborate on the role of oxide nanoparticles as theranostic tools in the fight against different diseases by citing and briefly discussing the following papers (DOI: 10.1016/j.molliq.2018.05.105, DOI: 10.1007/s00339-021-04759-4, DOI: 10.1016/j.molstruc.2018.04.016).

Response: The suggested papers were included and discussed in the Introduction section of the revised manuscript.

Reviewer 2 Report

The manuscript entitled "Could iron-nitrogen doping modulate the cytotoxicity of TiO2 nanoparticles?" examined the very important issue of TiO2 nanoparticles toxicity. TiO2 nanoparticles and nanoparticles co-doped with iron (1%) and nitrogen atoms are well characterized by using multiple relevant techniques to get more insight into the microstructural properties. Likewise, authors used multiple relevant cytotoxicity markers to get more insight into the mechanism of cytotoxicity of TiO2 nanoparticles and nanoparticles co-doped with iron and nitrogen atoms. Therefore, I suggest only several minor modifications of the manuscript prior the acceptance. This is a nice paper, I would like to congratulate the authors on their work.

  1. The magnifications in figure 3A,B,C,D and also figure 7A,B seem to need attention it is hard to judge and compare the cell viability (A) and LDH release (B) and organization of actin cytoskeleton (3C,D) and the autophagic vacuoles (7A,B) of lung fibroblasts. Moreover, there is only one scale bar! Authors should add a standard scale bars to all subimages.

  1. The authors should use µM, nM instead of µmole, nmole (section 2.6.).

Author Response

The manuscript entitled "Could iron-nitrogen doping modulate the cytotoxicity of TiO2 nanoparticles?" examined the very important issue of TiO2 nanoparticles toxicity. TiO2 nanoparticles and nanoparticles co-doped with iron (1%) and nitrogen atoms are well characterized by using multiple relevant techniques to get more insight into the microstructural properties. Likewise, authors used multiple relevant cytotoxicity markers to get more insight into the mechanism of cytotoxicity of TiO2 nanoparticles and nanoparticles co-doped with iron and nitrogen atoms. Therefore, I suggest only several minor modifications of the manuscript prior the acceptance. This is a nice paper, I would like to congratulate the authors on their work.

  1. The magnifications in figure 3A,B,C,D and also figure 7A,B seem to need attention it is hard to judge and compare the cell viability (A) and LDH release (B) and organization of actin cytoskeleton (3C,D) and the autophagic vacuoles (7A,B) of lung fibroblasts. Moreover, there is only one scale bar! Authors should add a standard scale bars to all subimages.

Response: Thank you very much for your careful revision of this article. We greatly appreciate all your appreciation and remarks. We used this magnification in order to show an overview of the cells. Therefore, we enlarged Figures 3 and 7 for better observation and comparison of the results in the revised manuscript. Also, scale bars were added to all subimages in the new Figures of the revised manuscript.

  1. The authors should use µM, nM instead of µmole, nmole (section 2.6.).

Response: Thank you very much for your suggestion but enzyme activity units are expressed in µmoles of substrate modified by 1 mL of enzyme in one minute at 25 degrees Celsius. Therefore, µM cannot be used to express enzymatic activity.
